# Electrostatic lateral interactions drive ESCRT-III heteropolymer assembly

**Sudeep Banjade[1,2], Shaogeng Tang[1,2†], Yousuf H Shah[1,2], Scott D Emr[1,2]\***

[1]Weill Institute for Cell and Molecular Biology, Cornell University, Ithaca, United States; [2]Department of Molecular Biology and Genetics, Cornell University, Ithaca, United States

**Abstract** Self-assembly of ESCRT-III complex is a critical step in all ESCRT-dependent events. ESCRT-III hetero-polymers adopt variable architectures, but the mechanisms of inter-subunit recognition in these hetero-polymers to create flexible architectures remain unclear. We demonstrate in vivo and in vitro that the *Saccharomyces cerevisiae* ESCRT-III subunit Snf7 uses a conserved acidic helix to recruit its partner Vps24. Charge-inversion mutations in this helix inhibit Snf7-Vps24 lateral interactions in the polymer, while rebalancing the charges rescues the functional defects. These data suggest that Snf7-Vps24 assembly occurs through electrostatic interactions on one surface, rather than through residue-to-residue specificity. We propose a model in which these cooperative electrostatic interactions in the polymer propagate to allow for specific inter-subunit recognition, while sliding of laterally interacting polymers enable changes in architecture at distinct stages of vesicle biogenesis. Our data suggest a mechanism by which interaction specificity and polymer flexibility can be coupled in membrane-remodeling heteropolymeric assemblies.
DOI: https://doi.org/10.7554/eLife.46207.001

**\*For correspondence:**
sde26@cornell.edu

**Present address:** [†]Department of Biochemistry, Stanford University, Stanford, United States

**Competing interests:** The authors declare that no competing interests exist.

## Introduction

Eukaryotic organelles that are demarcated by membranes undergo continuous remodeling to maintain their integrity and function. Different evolutionarily conserved heteropolymeric machines and scaffolds including ESCRTs (endosomal sorting complexes required for transport), dynamin, clathrin, COP coatmers, BAR-domain containing proteins, etc. control the remodeling at different organelles (*Zimmerberg and Kozlov, 2006*). This polymerization-mediated membrane remodeling in eukaryotes is important for various cellular events such as trafficking, cell-division and migration. In bacteria, structural homologs of actin and tubulin (that include FtsZ and MreB proteins) also form heteropolymers, which deform and remodel membranes, leading to cell shape maintenance and division (*Eun et al., 2015*). In most cases these proteins are recruited to the membrane from the cytosol as monomers, polymerization then occurs on the 2D membrane, which generates the mechanical force to bend membranes. Given the functional importance of polymerization, elucidating mechanisms of how these various machines recognize their partners to form heteropolymers, and how these polymers adapt to physical and mechanical changes in the cellular environment is critical to our overall understanding of how cells control membrane homeostasis.

ESCRTs drive an ever-growing list of cellular processes. These include biogenesis of multivesicular bodies (MVBs) (*Katzmann et al., 2001*; *Odorizzi et al., 1998*), viral budding (*Garrus et al., 2001*), cytokinesis (*Carlton and Martin-Serrano, 2007*), nuclear envelope reformation (*Olmos et al., 2015*; *Vietri et al., 2015*), plasma membrane repair (*Jimenez et al., 2014*), surveillance of defective nuclear pore complexes (*Webster et al., 2014*), lysosomal protein degradation (*Zhu et al., 2017*), endolysosomal repair (*Skowyra et al., 2018*), and so on *Hurley (2015)*.

This machinery consists of five evolutionarily conserved complexes, ESCRT-0, I, II, III and the AAA ATPase Vps4. ESCRT-0, I and II are recruited to the membrane through interactions with

**eLife digest** Cells are separated from the outside environment by a fatty layer called the plasma membrane. This layer not only isolates the inside of the cell from the outside, it is also essential for the cell to sense and respond to cues around it. For example, the plasma membrane contains different types of proteins that can act as receptors for signals from outside the cell or as channels to take in essential nutrients. One of the ways that the cell can respond to its environment is by recycling the proteins at the plasma membrane.

During a cell's life, proteins from its membrane are recycled by being pulled into lysosomes, which are sacs or vesicles full of enzymes that digest these molecules. However, before reaching the lysosomes, the molecules pass through another set of vesicles called endosomes. There, ESCRT-III, a flexible scaffold made out of the proteins Snf7, Vps24 and Vps2, forms a spiral like-structure that collects the proteins and fats from the membrane. This corkscrew-like shape allows the ESCRT-III scaffold to work, but it is unclear how it is formed.

Snf7 is a protein that forms long bending chains, or "polymers", by linking to itself. Banjade et al. found that Snf7 uses its negatively charged surface to interact with the parallel chain that Vps24 and Vps2 form at its side. However, Vps24 and Vps2 do not fit rigidly into Snf7 like a key fits in a lock. Rather, their interaction is flexible, based on charge. This flexibility may allow Vps24 and Vps2 to slide along the side of the Snf7 chain, helping to create a spiral. Banjade et al. used budding yeast as a model organism and also imaged purified proteins with electron microscopy to come upon these findings.

Understanding how ESCRT proteins interact to form complex structures may lead to a better understanding of how other membrane-bound polymers form elsewhere in the cell. ESCRT proteins are also involved in degenerative diseases, such as Alzheimer's, where proteins that need to be recycled cannot be properly processed, and they are important for viruses such as HIV to spread between cells. Understanding how these proteins interact to form their characteristic spiral structure could potentially lead to the development of new therapies.

DOI: https://doi.org/10.7554/eLife.46207.002

ubiquitinated proteins, while ESCRT-0 and ESCRT-II also bind phosphatidyl-inositol-3-phosphate (PI3P) at membranes (*Robinson et al., 1988*; *Rothman et al., 1989*; *Raymond et al., 1992*; *Katzmann et al., 2001*; *Babst et al., 1997*; *Katzmann et al., 2003*; *Babst et al., 2002a*; *Babst et al., 2002b*). ESCRT-II recruits ESCRT-III, also nucleating ESCRT-III polymerization. Polymerization of ESCRT-III is dynamically rearranged and directed towards disassembly via the recruitment of the AAA ATPase Vps4 (*Mierzwa et al., 2017*; *Adell et al., 2017*).

The requirement of early ESCRTs (0, I and II) varies between the different membranes. For example, in viral budding processes, the ESCRT-0 complex is not required, while in cytokinetic abscission neither ESCRT-0 nor ESCRT-II are required. ESCRT-III, however, is ubiquitously required in all ESCRT-dependent processes. This membrane-modifying property of ESCRT-III is assumed to be a direct consequence of its self-assembly into meso-scale polymers. The requirement of ESCRT-III function in all ESCRT-dependent processes emphasizes the importance of understanding the molecular details of ESCRT-III co-assembly.

ESCRT-III proteins are ~26 KDa charged proteins that assemble into a polymer at membranes. In yeast, the 'core' ESCRT-III machinery consists of Vps20, Snf7, Vps24 and Vps2, as the deletion of any of these four genes leads to a strong defect in the MVB pathway (*Babst et al., 2002b*). Other ESCRT-III proteins consist of Ist1, Did2, Vps60 and Chm7, which have accessory roles in the MVB pathway, but are essential in mammalian cytokinesis (Ist1/Did2) (*Bajorek et al., 2009a*) or nuclear envelope sealing (Chm7) (*Webster et al., 2016*).

All ESCRT-III proteins consist of an N-terminal helical bundle, which is predominantly basic, and a C-terminal flexible and acidic region (*Muziol et al., 2006*; *Bajorek et al., 2009b*). The N-terminal helical bundle is known to be important for ESCRT-III co-assembly (*Henne et al., 2012*), while the C-terminus recruits the AAA ATPase Vps4 (*Obita et al., 2007*).

Different architectures of ESCRT-III polymers have been visualized with electron microscopy – flat sheets, rings, spirals and helices, owing to the flexible nature of the polymers (*Henne et al., 2012*;

*Hanson et al., 2008*; *Chiaruttini et al., 2015*; *Shen et al., 2014*). The most abundant of the ESCRT-III proteins, Snf7, forms 2D spirals. Combination of Snf7 with downstream proteins Vps24 and Vps2 changes the architecture of these spirals to form three dimensional helical polymers (*Henne et al., 2012*). The mammalian orthologues of Vps24 and Vps2 (named CHMP3 and CHMP2) co-assemble to form helical structures (*Lata et al., 2008*), whereas Vps24 has been visualized to form linear polymers (*Ghazi-Tabatabai et al., 2008*).

Despite our understanding of the individual homo-polymerization mechanism (Snf7-Snf7), the mechanisms of how ESCRT-III proteins recognize one another to create heteropolymers remain unclear. While recruitment of downstream ESCRT-III proteins Vps24 and Vps2 by Snf7 is established, what roles these partners of Snf7 play in regulating the polymers of Snf7 remain unclear. How these proteins change the architecture of the flat 2D spirals of Snf7 to 3D helical structures also remains undefined.

Furthermore, in analyzing ESCRT-III polymerization, most studies only focus on either biochemical/structural approaches, or only cell biological approaches, with a limited number of studies combining both approaches to understand ESCRT-III function in vivo. Here, we apply a combination of in vivo and in vitro approaches to shed light on the hetero-polymerization mechanisms of ESCRT-III proteins Snf7 and Vps24. In particular, we utilize powerful directed evolution-based approaches in yeast to select for Snf7 mutants that rescue polymerization defects. These approaches allow us to describe in vivo mechanisms that ESCRT-III utilizes to co-assemble in the MVB pathway and provide guidance to understand existing literature on ESCRT-III filament structures.

We previously reported the crystal structure of Snf7 (*Tang et al., 2015*), which depicted the ESCRT-III subunit in an 'active,' polymerization-capable conformation. This structure provided important clues regarding how the polymerization of Snf7 is regulated. Here, we identify and characterize a motif in Snf7 critical for recruitment of its downstream partner Vps24. Mutagenesis analyses suggest that this motif is involved in Snf7/ESCRT-III lateral interaction in vitro. The acidic nature on its solvent-exposed surface, rather than a consensus sequence, is important for the function of this helix, suggesting an electrostatic mechanism that Snf7, and most likely all ESCRT-III subunits, use to recognize their partners. We hypothesize that ESCRT-III proteins utilize these lateral charge interactions with weak residue-to-residue specificity to slide on the filaments, enabling them to form spiraling polymers of different diameters and architectures.

## Results

### Mutations in the peripheral helix-4 of Snf7 induce cargo-sorting defects

Our structural analysis of Snf7 allowed us to specify helices 1, 2 and 3 as the core polymeric interface that drives longitudinal polymerization. Structure/function studies of the *Drosophila* Snf7 (Shrub) confirmed the existence of this interface in the polymer, showing nearly identical packing arrangement in the polymer (*Tang et al., 2015*; *McMillan et al., 2016*), despite some differences in the side-chain residues. Additionally, the structure of the similarly organized CHMP1B in its helical assembly with IST1 suggests that the same core interface drives ESCRT-III polymerization with an evolutionarily conserved mechanism of ESCRT-III assembly (*McCullough et al., 2015*; *McCullough et al., 2018*; *Talledge et al., 2019*).

Helix-4 of Snf7 lies at the periphery of this core longitudinal interface (*Figure 1A–1B*) and stretches from residue ~120 to~150. In the crystal structure of Snf7$^{core}$, which included residues 12–150, we observed that residues D124 to E138 are structured, while the rest of the amino acids are not visible. Helix-4 is mostly acidic in nature, with the acidic residues falling on one interface (*Figure 1A*, *Figure 1—figure supplement 1A*).

We had previously observed that deleting helix-4 of Snf7 resulted in a defect in recruiting Vps24-GFP to endosomes, although we did not have a mechanistic understanding behind this phenotype (*Henne et al., 2012*). Interestingly, the previously solved cryo-EM structure of CHMP1B (which is the mammalian homolog of Did2) depicted extensive electrostatic contacts made by CHMP1B's acidic residues of helices 4 and 5 with the basic helix-1 of IST1 (*McCullough et al., 2015*). The overall acidity of Snf7 helix-4 is conserved in the Snf7 orthologues of *Saccharomyces cerevisiae*, *Homo sapiens*, *Mus musculus*, *Xenopus laevis*, *Drosophila melanogaster*, *Caenorhabditis elegans*, and

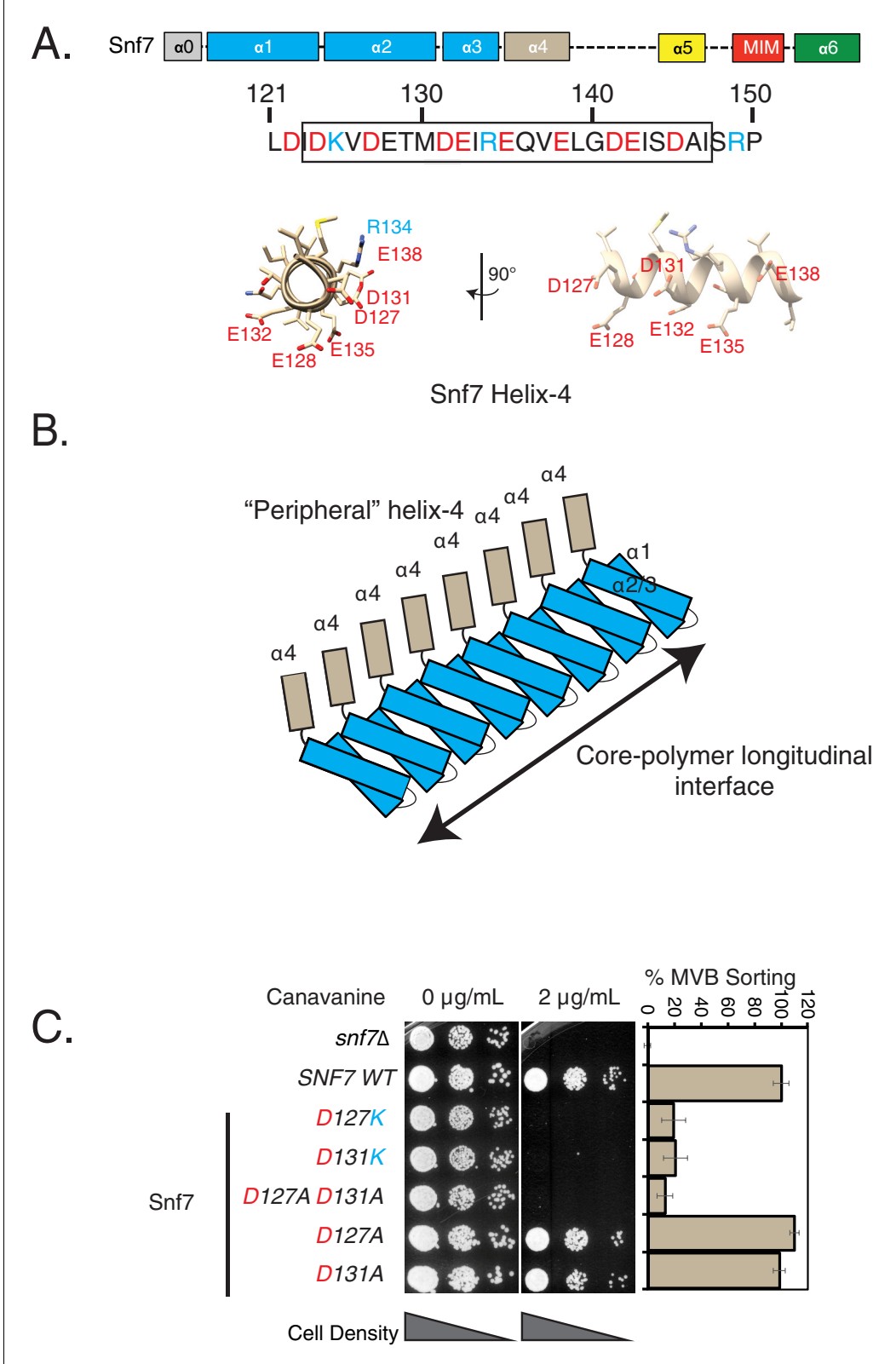

**Figure 1.** Mutations in helix-4 region of Snf7 induce cargo-sorting defect. (**A**) Domain organization of Snf7, depicting the different helices (top). Bottom figure shows the sequence of helix-4, with the predicted helical motif highlighted with a box. Acidic residues are denoted in red, while cyan residues are basic amino acids. Bottom – structure of the helix-4 (from PDB 5fd7) in two orientations, highlighting the acidic residues on one surface. (**B**) Cartoon model of the polymeric arrangement of Snf7 in its linear form observed in the crystal lattice. (**C**) Canavanine sensitivity and Mup1-pHluorin flow-

*Figure 1 continued on next page*

*Figure 1 continued*

cytometry data (right) showing cargo-sorting/endocytosis defects of the helix-4 mutants of Snf7. Mup1-pHluorin data were collected 90 min after methionine addition. Error bars represent standard deviation from 3 to 7 independent experiments.

DOI: https://doi.org/10.7554/eLife.46207.003

The following source data and figure supplements are available for figure 1:

**Source data 1.** Individual data points for data plotted in the figure (*Figure 1C*) for % MVB sorting.
DOI: https://doi.org/10.7554/eLife.46207.008
**Figure supplement 1.** Snf helix-4 consists of conserved acidic residues.
DOI: https://doi.org/10.7554/eLife.46207.004
**Figure supplement 2.** Models of the cargo-sorting assays used in this study using *Saccharomyces cerevisiae*.
DOI: https://doi.org/10.7554/eLife.46207.005
**Figure supplement 3.** Mutational analysis of helix-4 residues of Snf7.
DOI: https://doi.org/10.7554/eLife.46207.006
**Figure supplement 4.** Helix-4 mutation does not affect stability of Snf7.
DOI: https://doi.org/10.7554/eLife.46207.007

*Schizosaccharomyces pombe* (*Figure 1—figure supplement 1B*), which suggests an important role of the electrostatics mediated by these charged residues.

Based on these analyses and the insights provided by the recent atomic structures, we hypothesized that the acidic interface of helix-4 may be important for Snf7's assembly with its other ESCRT-III partners.

We comprehensively analyzed this acidic interface of helix-4 using endocytosis assays in *Saccharomyces cerevisiae*. One of the well-developed assays utilizes canavanine, a toxic analog of arginine. ESCRT mutant strains are sensitive in their growth in the presence of the drug, as the ESCRT mutants are incapable of downregulating the canavanine transporter Can1 from the plasma membrane (*Lin et al., 2008*) (*Figure 1—figure supplement 2A*).

Using this canavanine-sensitivity assay, we found that mutations in the acidic residues of helix-4 have strong sensitivity to canavanine (*Figure 1C*, *Figure 1—figure supplement 3A*). Mutations in residues D131 and D127 show the strongest phenotype (*Figure 1C*). Charge-inversion single mutations (D to K) or double-alanine (D127A D131A) mutations showed strong effects in cargo sorting, while single mutations to alanine do not show any defects (*Figure 1C*).

In an orthogonal ESCRT dependent cargo-sorting assay, these helix-4 mutants show a similar behavior. We have routinely used the sorting and degradation of the methionine transporter Mup1 fused to the pH sensitive pHluorin to analyze ESCRT mutants (*Henne et al., 2012*; *Tang et al., 2016*)(*Figure 1—figure supplement 2B*). The defect of the helix-4 mutants of Snf7 is consistently observed in the methionine-dependent sorting of Mup1-pHluorin (*Figure 1C* and *Figure 1—figure supplement 3C*).

In our assays, we observed that mutations in residues D127, D131, E138 and E142 gave the strongest effects (*Figure 1—figure supplement 3C*). Importantly, D127, D131 and E138 lie on the same surface on the structure of helix-4 (*Figure 1A*, *Figure 1—figure supplement 3C*).

The effects of the helix-4 mutants that we have observed are not due to the instability of the mutated proteins. In in vitro assays using the mutation D131K, the protein behaves similarly to the non-mutated version. We have previously analyzed the activating mutation in Snf7 (R52E), as the mutation lowers the critical concentration for polymerization (*Henne et al., 2012*). This mutation allows us to observe polymers of Snf7 on electron microscopy grids and on lipid monolayers. The helix-4 mutant D131K (with R52E) is able to form polymers of Snf7 (*Figure 1—figure supplement 4A*). One important exception here was that the D131K mutant showed defects in its lateral interaction, as it preferentially makes thinner filaments (*Figure 1—figure supplement 4A* -inset). A similar fraction (~50%) of Snf7 bound to liposomes with both Snf7$^{R52E}$ and Snf7 $^{R52E\ D131K}$ proteins (*Figure 1—figure supplement 4B*). Furthermore, both proteins showed a similar size-exclusion chromatogram (*Figure 1—figure supplement 4C*). These analyses suggest that the helix-4 mutation does not adversely affect folding of Snf7.

## Helix-4 mutations in Snf7 inhibit its interaction with Vps24

In the crystal lattice of Snf7$^{core}$, the helix-4 residue D131 makes electrostatic contacts with helix-1 residues K21 and K25 in trans on a laterally interacting polymer strand (*Figure 2—figure supplement 1A*). In vivo, mutating K21 and K25 individually do not have any defect in cargo sorting, while the double mutation *K21E K25E* has a mild defect (~60% MVB function (*Figure 2—figure supplement 1B–C*). Although these data do not rule out the possibility that this interface is involved in Snf7-Snf7 lateral interaction, further analyses below suggest that this is an interface mimicking Snf7's interaction with its partner Vps24.

The requirement of Snf7 to recruit Vps24 at endosomes has been well documented in the literature (*Teis et al., 2008*), and the in vitro binding of these proteins has also been documented (*Mierzwa et al., 2017*; *Henne et al., 2012*). However, the molecular description of these interactions is incomplete. We provide several lines of evidence that Snf7 uses helix-4 to recruit Vps24.

First, we found that helices 1–4 of Snf7 can recruit VPS24-GFP to endosomes. However, deleting helix-4 or making specific mutations in helix-4 (D131K) reduced the recruitment (*Figure 2A*, *Figure 2—figure supplement 2A*). In subcellular fractionation experiments, the helix-4 mutant snf7$^{D131K}$ recruited ~5 fold lower amount of Vps24 than the wild-type protein in the P13 (endosome-enriched) fraction (*Figure 2—figure supplement 2B*). Similarly, in co-immunoprecipitation experiments, the amount of Snf7$^{D131K}$ bound to Vps24 was reduced by >10 fold compared to wild-type (*Figure 2B*). Furthermore, in in vitro lipid monolayer assays, the Snf7$^{R52E\ D131K}$ mutant was unable to form 3D helical spirals with Vps24 and Vps2 (*Figure 2C*).

To further analyze ESCRT-III polymerization in vivo, we used rate-zonal velocity gradient assays. In these assays, we found that the Snf7 helix-4 mutant phenocopies the *vps24Δ* strain. We previously observed that in cells lacking Vps24, Snf7 sediments to fractions containing higher percent glycerol, indicating formation of higher order polymers of Snf7 (*Teis et al., 2008*). This phenomenon most likely occurs because of the inability of *vps24Δ* to recruit the AAA-ATPase Vps4 to endosomes, eliminating Vps4-mediated disassembly of ESCRT-III polymers. In the helix-4 mutant snf7$^{D131K}$ strain, we observed a strikingly similar phenotype, as the Snf7$^{D131K}$ protein sediments towards the bottom of the gradient, indicating that snf7$^{D131K}$ VPS24 phenocopies SNF7 vps24Δ (*Figure 2D*).

Altogether, these experiments provide strong evidence that the Snf7 helix-4 mutant is defective in recruiting Vps24.

## Basic to acidic mutations in helix-1 of Vps24 rescue the defect of Snf7 helix-4 mutants

To identify parameters that can overcome the defect of Snf7 helix-4 mutations and therefore to characterize the interacting surface of Snf7 on Vps24, we performed unbiased mutagenesis of Vps24 to look for suppressors of snf7$^{D131K}$. The snf7$^{D131K}$ mutant strain is canavanine sensitive, as ESCRT mutants are defective in endocytosis of Can1 (*Figure 1—figure supplement 2A*). Taking advantage of the canavanine sensitivity of snf7$^{D131K}$, we used random mutagenesis to select mutants of *vps24* that could rescue the defect of the Snf7 helix-4 mutation.

This selection approach identified several mutations on the helix-1 surface of Vps24 (Q16E, K26E and K33E of Vps24) that were canavanine resistant in the snf7$^{D131K}$ background. Of these, Q16E mutation on Vps24 gave the strongest effect in suppressing the defect of snf7$^{D131K}$, as ~70% of Mup1-pHluorin is sorted with the vps24$^{Q16E}$ mutation in the background of snf7$^{D131K}$ (*Figure 3A*, *Figure 3—figure supplement 1A–1C*). Interestingly, Vps24 mutants Q16E and R19E additionally also suppress the defects of other Snf7 mutations D127K and D142K (*Figure 3A*). Importantly, the expression levels of the Vps24 mutant proteins were similar to wild-type Vps24 (*Figure 3—figure supplement 1B*).

To study whether the residues of helix-1 of Vps24 lie in the vicinity of helix-4 of Snf7, we used ex vivo crosslinking approaches to crosslink these two proteins. Consistent with the idea that Snf7 helix-4 interacts with the helix-1 surface of Vps24, the sulfhydryl crosslinker BMOE crosslinks Snf7$^{D131C}$ to cysteine mutations in helix-1 of Vps24 (Vps24$^{Q16C}$, Vps24$^{R19C}$ and Vps24$^{K26C}$) (*Figure 3B*, *Figure 3—figure supplement 2B*). In these assays, we also see Snf7D131C:Snf7D131C crosslinks probably due to the fact that Snf7 spirals are flexible enough to bend and form crosslinks through the longitudinal surface (*Figure 3—figure supplement 2A–B*). The amount of crosslinked Vps24 decreases with additional mutations in the vicinity of D131 in Snf7 that reduce Vps24 binding

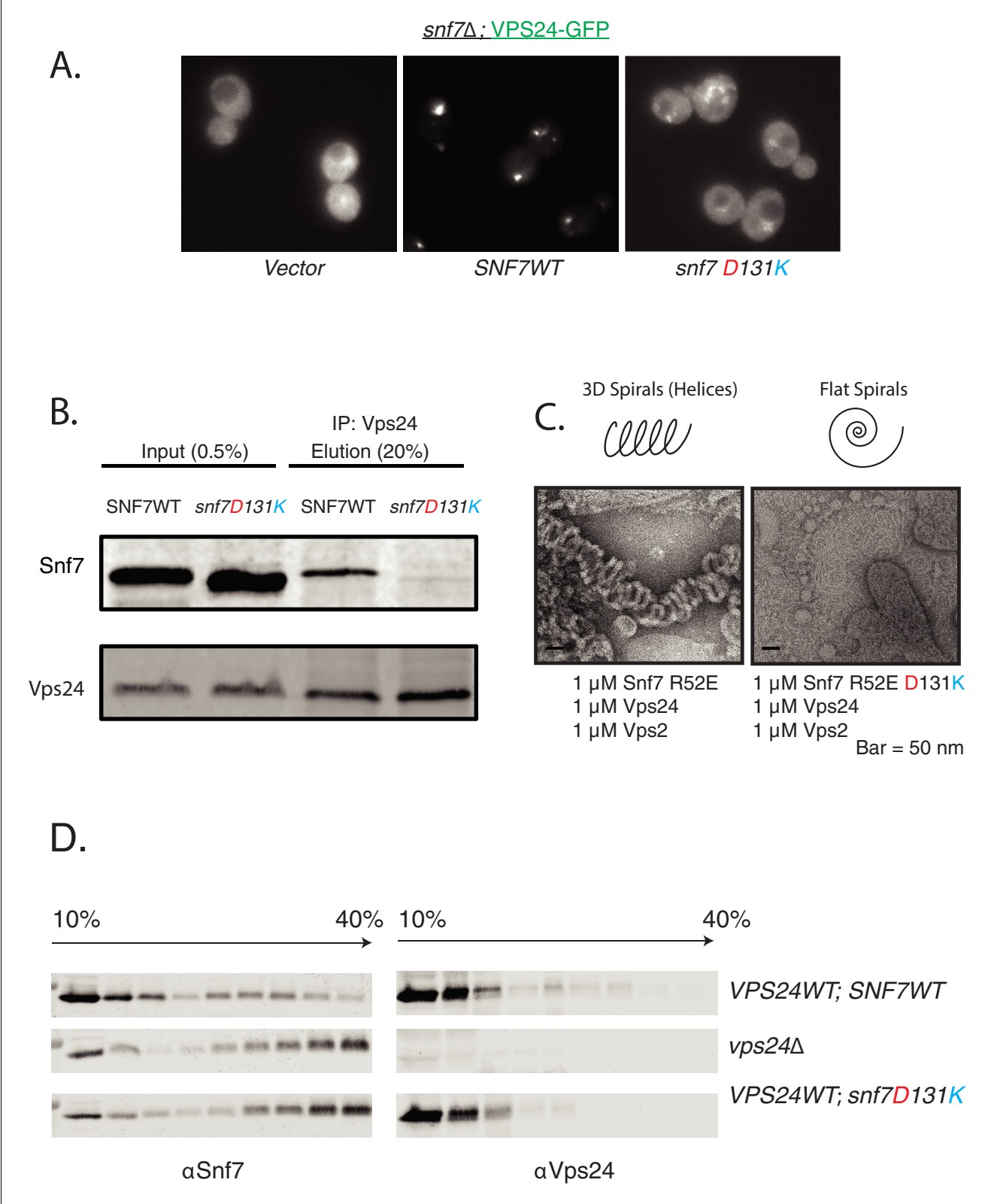

**Figure 2.** Helix-4 region of Snf is involved in recruiting Vps24. (**A**) Helix-4 mutants fail to recruit Vps24-GFP. Vps24-GFP is present in endosomal dots in cells expressing *SNF7 WT* on a plasmid in a *snf7Δ;Vps24-GFP* strain, while with the mutation on Snf7, Vps24-GFP is more diffuse in the cytoplasm. (**B**) Coimmunoprecipitation experiments, immunoprecipitating Vps24 and blotting for either wild-type Snf7 or the mutant D131K. (**C**) Electron microscopy assay on lipid monolayers, depicting co-assembly of Snf7 R52E with Vps24 and Vps2 into helices. Snf7 R52E D131K fails to form similar structures.
*Figure 2 continued on next page*

*Figure 2 continued*

Experiments were done with 1 µM of each protein incubated for 1 hr on lipid monolayers. (D) 'In vivo' glycerol-gradient experiments using a gradient of 10% to 40%, using various mutant strains annotated on the far-right. Western-blots were performed against Snf7 (left) or Vps24 (right).

DOI: https://doi.org/10.7554/eLife.46207.009

The following figure supplements are available for figure 2:

**Figure supplement 1.** Helix-4 contacts a *trans* Snf7 polymer in the crystal lattice.

DOI: https://doi.org/10.7554/eLife.46207.010

**Figure supplement 2.** Helix-4 region of Snf7 is involved in recruiting Vps24.

DOI: https://doi.org/10.7554/eLife.46207.011

**Figure supplement 3.** CHMP1B helix-4 contacts the helix-1 of IST1.

DOI: https://doi.org/10.7554/eLife.46207.012

(adding the D127K and E142K mutations - Snf7 $^{D131C\ D127K}$ and Snf7 $^{D131C\ D127K\ E142K}$) (**Figure 3— figure supplement 2A**). With the presence of these D127K and E142K mutants, we see stronger Snf7-Snf7 crosslinking, consistent with the observation that in the absence of Vps24, Snf7 forms higher amounts of polymers (**Figure 2D**).

Comparison of our data with the cryo-EM structure of CHMP1B and IST1 (**McCullough et al., 2015**) provides us with an additional support of the model of Snf7 helix-4 contacting Vps24 helix-1. In the solved structure (**Figure 2—figure supplement 3A–C**), the helix-4 residues E106, S109 and D113 of CHMP1B are in close proximity to form salt bridges with IST1 helix-1 residues N14 and R16. The polar and acidic residues of helix-4 in CHMP1B and in Snf7 are also conserved (**Figure 2—figure supplement 3C**). Additionally, the IST1 structure is similar to the homology model of Vps24 (using the hVps24 crystal structure as the template) (**Figure 2—figure supplement 3D**) implying that the helix-1 residues probably are similarly positioned in a copolymer.

Overall, the specific point mutations on Vps24 that suppress the defect of Snf7 helix-4 mutations also provide additional evidence that helix-4 of Snf7 is involved in recruiting its partner Vps24. Helix-4 on Snf7 is located at the periphery of the core polymer, indicating that Vps24 binds at the sides of the core polymer of Snf7. The location of the suppressor mutations on Vps24 also points to helix-1 of Vps24 as the interaction surface. These data are supported by the crosslinking assay, and the comparison with an orthogonal ESCRT-III assembly structure of CHMP1B and IST1 (**McCullough et al., 2015**). The suppression of separate residues D127 and E142 in Snf7 that are >21 Å apart (D127 and E138 are 21 Å apart and residues beyond E138 are not visible in the structure) by Vps24 helix-1 residues indicated an interacting surface that possesses weaker specificity.

## Charge inversion mutations in helix-4 are genetic suppressors of cargo-sorting defects

Through mutagenesis approaches, we have previously been able to identify several Snf7 activating mutations that allowed us to decipher how Snf7 is activated by upstream nucleating factors (**Tang et al., 2016**). We hypothesized that we would be able to similarly identify parameters in Snf7 that would enhance the affinity of Snf7 to Vps24, using the *snf7$^{D131K}$* mutant that has defects in Vps24 binding. Therefore, we additionally performed error-prone mutagenesis with Snf7 to look for intragenic suppressors of D131K. We mutagenized *snf7$^{D131K}$* on a plasmid and selected for mutants that rescue the canavanine-sensitivity phenotype.

Through this genetic selection approach, we found that several additional mutations in helix-4 of Snf7 that balanced the acidity of the helix rescued the defect of the D131K mutation. As described above, charge-inversion mutations on the helix-1 surface of Vps24 rescued the defect of the Snf7 helix-4 mutations. Consistent with this, the following pieces of data suggest that rescuing the acidic defect of the D131K mutation rescues functional phenotype of the Snf7 helix-4 mutations.

We found that charge-inversion double mutations D131K R134D and D131K R149D in the helix-4 region of Snf7 completely sorts Mup1-pHluorin (~100%), unlike the defective single mutant D131K (**Figure 4A–4B**, **Figure 4—figure supplement 1C**). Additionally, the double mutant D127A D131A was also rescued by the additional inclusion of the mutations R134D or R149D (**Figure 4—figure supplement 1A and C**).

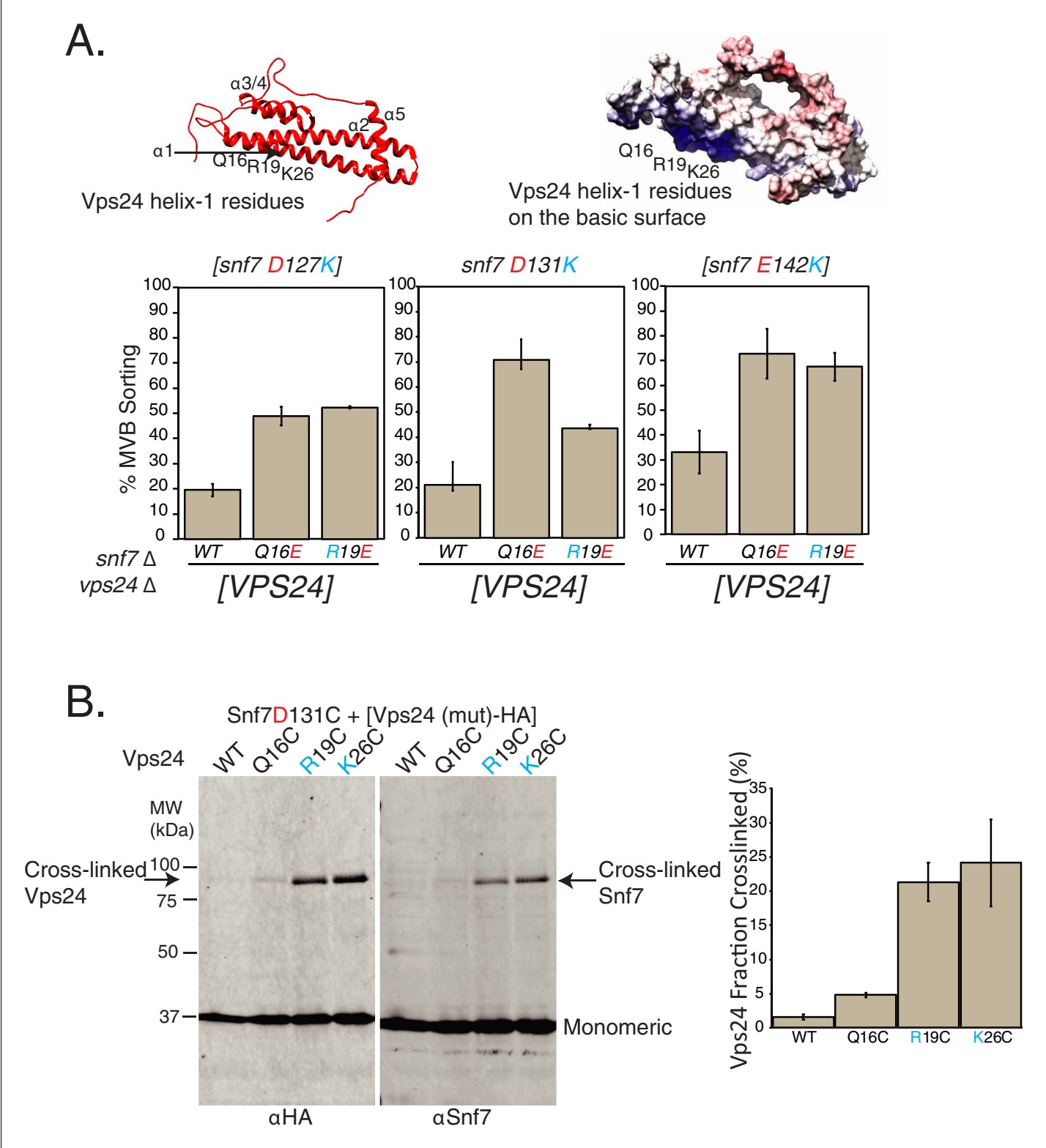

**Figure 3.** Random mutagenesis selection identified suppressors of Snf7-helix-4 mutations on Vps24. (**A**) Locations of random mutations on the homology model of Vps24 (top), obtained through canavanine-resistance selection with mutagenized Vps24. The crystal structure of CHMP3 (PDB 3FRT) was used to create the homology model. Figure on the top-right shows the electrostatics of the same model. Graphs below represent Mup1-pHluorin assays, where the mutations in helix-1 of Vps24 rescue the defects of the indicated Snf7 mutants. Error bars represent standard deviation from three to five independent experiments. (**B**) Ex vivo BMOE mediated crosslinking assays between cysteines in Snf7 and Vps24 at the indicated positions.

*Figure 3 continued on next page*

Figure 3 continued

Immunoblots were performed against Snf7 or HA (on Vps24). Arrows indicate crosslinked species. Quantification on the right represents three independent replicates of the fraction of Vps24 (mutant) crosslinked with Snf7 D131C.

DOI: https://doi.org/10.7554/eLife.46207.013

The following source data and figure supplements are available for figure 3:

**Source data 1.** Individual data points for data plotted in the figure (*Figure 3A*) for % MVB sorting.

DOI: https://doi.org/10.7554/eLife.46207.016

**Source data 2.** Data plotted in the figure for % Vps24-HA crosslinked in *Figure 3B*.

DOI: https://doi.org/10.7554/eLife.46207.017

**Figure supplement 1.** Identification of the surface of Vps24 that interacts with Snf7 helix-4.

DOI: https://doi.org/10.7554/eLife.46207.014

**Figure supplement 2.** Crosslinking analyses of Snf7 and Vps24.

DOI: https://doi.org/10.7554/eLife.46207.015

Upon further analysis, to our surprise, mutating the basic residue R134 or R149 to alanine also rescued the defect of the D131K or the double-mutant D127A D131A (*Figure 4—figure supplement 1A*). These observations hold true for both model cargos Can1 and Mup1.

Our data suggest that maintaining the acidic nature of one surface of helix-4 that is important for Vps24 recruitment rescues the defective phenotype. In the structure of Snf7 helix-4, K125 points away from the acidic region (*Figure 4A*). Mutating this Lys (K125) to Glu in *snf7*$^{D131K}$ only partially suppressed the defect (*Figure 4—figure supplement 1A*), unlike the charge-inversion mutants of the same surface of the helix (R134, R149). Mutations in the polar amino acid Q136 to Glu on the opposite surface of the helix did not suppress the defect of *snf7*$^{D131K}$ (*Figure 4—figure supplement 1A*).

These charge-inversion suppression effects were also observed with the defects of other mutations in helix-4 of Snf7 - E142K (*Figure 4—figure supplement 1B*) and that of D127K (*Figure 4—figure supplement 1D*). Mutations R134 to Glu, or R149 to Glu completely suppressed the defects of both D127K and E142K.

These data strongly argue that the overall acidic nature of the helix-4 surface of Snf7 is important for its function. Following the observations that mutations in helix-4 affect binding to Vps24, the overall acidity of helix-4 appears to mediate recruitment of Snf7's partner Vps24 to the polymer.

## Lateral cooperative interactions mediated by electrostatics drive ESCRT-III co-assembly

ESCRT-III assemblies are unique biological polymers, as they form membrane-bound spirals of different architectures that achieve a topologically unique form of membrane bending. Snf7 alone primarily forms flat spirals in vitro (*Henne et al., 2012*; *Chiaruttini et al., 2015*; *Shen et al., 2014*). Overexpression of CHMP4, the mammalian homologue of Snf7 produces flat spirals or 3D helical structures in cells, that are similar to the spirals observed in vitro (*Hanson et al., 2008*).

We previously observed that incubating Snf7$^{R52E}$, Vps24 and Vps2 on lipid monolayers produce architecturally distinct polymers similar to three-dimensional super-helical structures (*Henne et al., 2012*), distinct from the flat, 2D spirals. Upon further analysis in this current study, we find a strong thermodynamic and kinetic dependence on the formation of these helices. At lower concentrations (1 µM each of Snf7$^{R52E}$, Vps24 and Vps2) and shorter incubation times on lipid monolayers (10 min), all three proteins are required for the formation of 3D helices (*Figure 5—figure supplements 1–2*). However, at higher concentrations (7 µM of Snf7$^{R52E}$) and shorter times (10 min), just Snf7$^{R52E}$ and Vps24 can together form 3D helices, albeit at a lower frequency (*Figure 5—figure supplement 2A and C*). We occasionally observe 3D helices of Snf7$^{R52E}$ alone at higher concentrations (7 µM) and longer incubation times of 60 min (*Figure 5—figure supplement 2B*), but these occur at a very low frequency (*Figure 5—figure supplement 2C*).

These data suggest that Snf7 has an intrinsic flexibility to transform into 3D helices, which are accelerated by the addition of its partners Vps24/Vps2. Similar to this property of Snf7, all ESCRT-III subunits likely have an ability to form flexible polymers, the architecture of which can be modulated by various properties – conformational changes, lateral strain (*Chiaruttini et al., 2015*), surface density of the filaments and crowding effect of proteins (cargo/filaments).

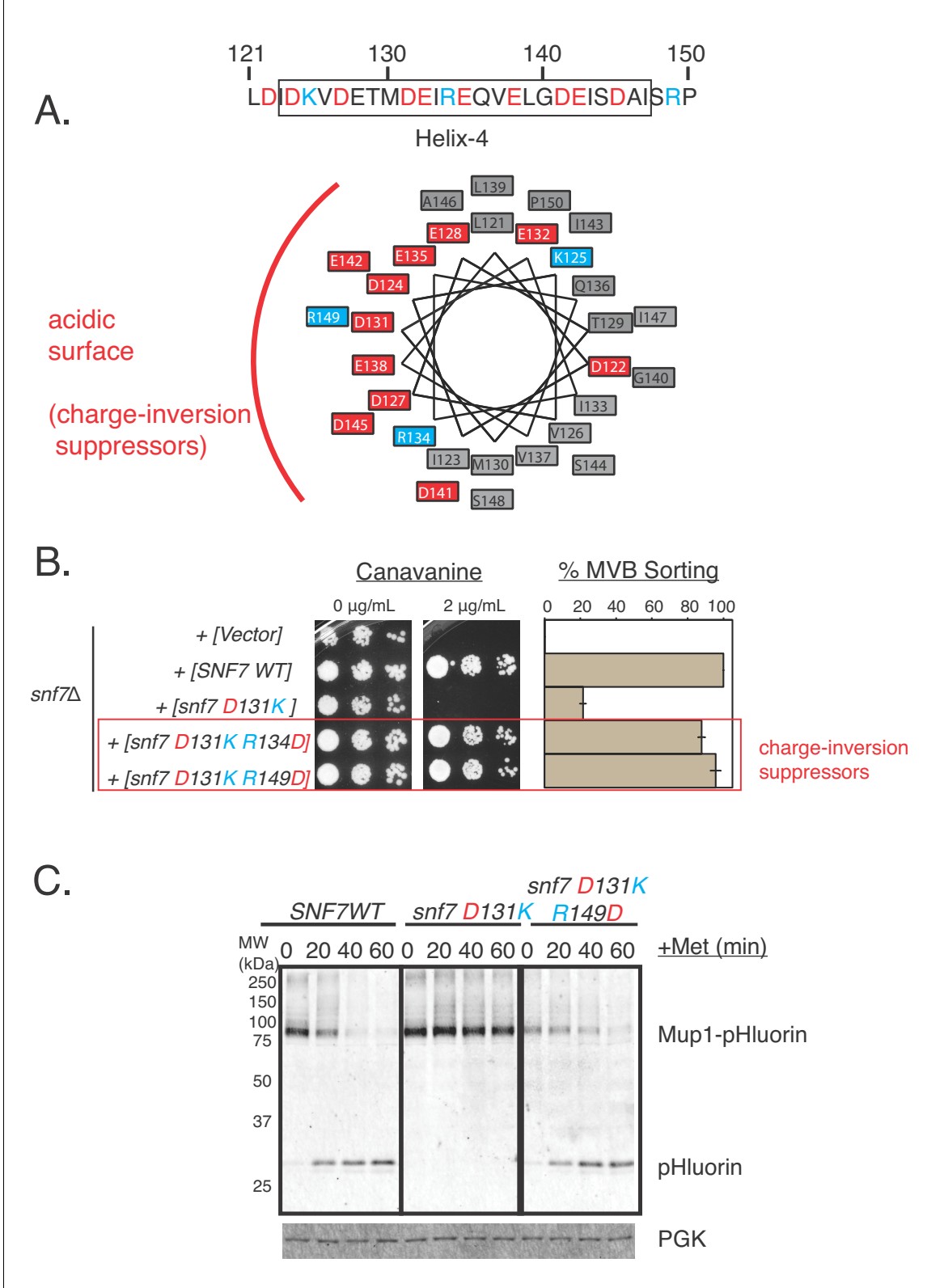

**Figure 4.** Random mutagenesis approach discovered charge-inversion suppressors in helix-4 of Snf7. (**A**) Helical-wheel representation of the helix-4 region of Snf7, highlighting acidic (red) and basic (cyan) residues. (**B**) Canavanine-sensitivity (left) and Mup1-pHluorin assays (right). Opposite charge mutations (positive to negative charges) rescue cargo-sorting defects of Snf7 mutants. Error bars represent standard deviation from three independent

*Figure 4 continued on next page*

*Figure 4 continued*

experiments. (C) Similar experiments as in (B - right) showing Mup1-pHluorin degradation over time after addition of methionine with mutants *snf7 D131K* and *snf7 D131K R149D*, immunoblotting for pHluorin.

DOI: https://doi.org/10.7554/eLife.46207.018

The following source data and figure supplement are available for figure 4:

**Source data 1.** Individual data points for the %MVB sorting of Snf7 mutant and the charge suppressor in *Figure 4B*.

DOI: https://doi.org/10.7554/eLife.46207.020

**Figure supplement 1.** Charge-inversion in helix-4 rescue cargo-sorting defects of mutants.

DOI: https://doi.org/10.7554/eLife.46207.019

Pure filaments of Snf7 also have an ability to form lateral interactions into bundles, as we and others have observed before (*Henne et al., 2012*; *Chiaruttini et al., 2015*). This lateral interaction of Snf7 polymers also seems to depend upon both the kinetics and thermodynamics of polymerization. At lower concentrations (500 nM) and shorter incubation time (10 min), Snf7$^{R52E}$ can form single stranded filaments (*Figure 5—figure supplement 3*). At higher concentrations (1 μM and above), this mutant predominantly forms bundles through lateral interactions (*Figure 5—figure supplement 3*). Therefore, depending upon the exact stage of polymerization, Snf7 (and likely other ESCRT-III proteins) can form several laterally associating strands.

The activation mutant R52E predominantly forms laterally interacting filaments of Snf7 as its critical concentration of polymerization is appreciably lowered. The difference in thickness of polymers of Snf7$^{R52E}$ with and without Vps24/Vps2 is not easily apparent, especially as this condition represents later stages of polymerization. At these later stages of polymerization, which occur after activation of Snf7, the polymers may consist of helices that are capable of inducing membrane deformation.

To observe the earlier stages of copolymer assembly, in our assays, we used wild-type Snf7 in the presence of Snf7's nucleator Vps20. With these assays, we observed predominantly single-stranded filaments of Snf7 (width of ~4–5 nm) (*Figure 5*). Under these conditions of earlier stages of polymer assembly, Snf7 filaments also do not form complete spirals (*Figure 5*). With the addition of Vps24 and Vps2, thicker filaments were formed, suggesting that Vps24 and Vps2 induce formation of laterally-interacting polymers, consistent with previous observations (*Mierzwa et al., 2017*). Importantly, the helix-4 mutant Snf7$^{D131K}$ is defective in inducing lateral bundles (*Figure 5*) with Vps24/Vps2, while the suppressor Vps24$^{Q16E}$ mutant forms a higher density of polymers and also partially rescues the bundling property.

With wild-type Snf7, even with the inclusion of the nucleator Vps20, we do not observe helices of Snf7-Vps24-Vps2. With the activated mutant Snf7$^{R52E}$, along with Vps24 and Vps2, the presence of Vps20 does not inhibit helix formation (*Figure 5—figure supplement 4*). These data suggest that Vps20 alone is insufficient to fully activate Snf7 in our in vitro assays. This most likely occurs because our in vitro protein Vps20 is not myristoylated, as it is normally in vivo (*Teis et al., 2010*). Snf7 is activated by Vps20, and also by ESCRT-II (*Henne et al., 2012*). Adding an additional component in our assay – an ESCRT-II component (GST tagged Vps25), induces helicity (*Figure 5—figure supplement 4*), consistent with the idea that a higher level of activation of Snf7 in the in vitro assays is necessary to produce helices.

Our data are most consistent with the model that Vps24 and Vps2 can laterally associate with Snf7 filaments to induce bundles of ESCRT-III at earlier times and lower activation threshold. Over time and at later stages of polymerization, ESCRT-III filaments mold into 3D helices that are more likely to be the mature polymers, structures that can generate the mechanical force important for vesicle-budding reactions. The lateral interactions are promiscuous and likely flexible enough for the polymers to be able to constrict into different architectures.

## In vivo analyses of cooperative assembly of Snf7, Vps24 and Vps2

Our in vitro data imply that Vps24 and Vps2 cooperatively bind to the Snf7 filament, inducing bundling and architectural changes in the polymer. To analyze cooperative assembly of Snf7, Vps24 and Vps2 in vivo, we utilized the property of helix-4 mutants' cargo-sorting defects.

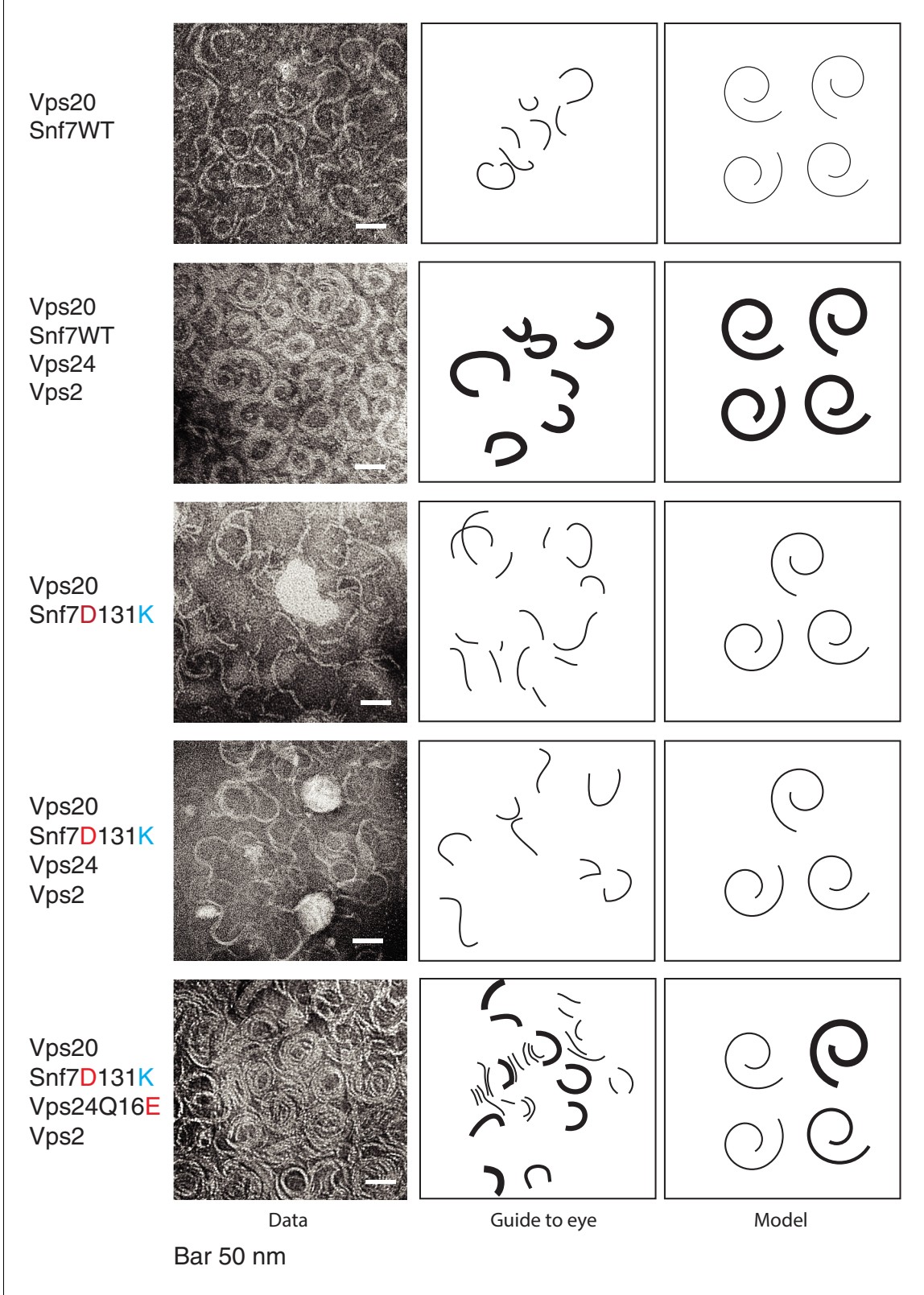

**Figure 5.** Vps24/Vps2 form lateral copolymer with Snf7. (**A**) Electron microscopy analysis of 10 μM Vps20, 1 μM Snf7 or (Snf7 D13K), 1 μM Vps24 (or Vps24 Q16E) and 1 μM Vps2. Experiments were done on lipid monolayers, with incubation of proteins for 1 hr. Painted lines on the middle-panel are guide to the eye to demonstrate differences in the thickness of the copolymers. The right panel shows hypothetical model of the structures of the polymers made from the different proteins.

*Figure 5 continued on next page*

*Figure 5 continued*

DOI: https://doi.org/10.7554/eLife.46207.021

The following source data and figure supplements are available for figure 5:

**Figure supplement 1.** Vps24/Vps2 form 3D helices at higher activation conditions of Snf7.

DOI: https://doi.org/10.7554/eLife.46207.022

**Figure supplement 2.** Snf7/Vps24/Vps2 form 3D helices at higher activation conditions.

DOI: https://doi.org/10.7554/eLife.46207.023

**Figure supplement 2—source data 1.** Snf7/Vps24/Vps2 helices formation is higher when all three proteins are present.

DOI: https://doi.org/10.7554/eLife.46207.024

**Figure supplement 3.** Snf7 filament progresses into lateral bundles over time.

DOI: https://doi.org/10.7554/eLife.46207.025

**Figure supplement 4.** Snf7 activation allows ESCRT-III helix formation.

DOI: https://doi.org/10.7554/eLife.46207.026

We reasoned that if simply the affinity of Vps24 to Snf7 is reduced in the charge-inversion mutants of helix-4 in Snf7, we would be able to partially restore this interaction, and the in vivo defects in cargo-sorting, by over-expressing Vps24.

Since Vps24 and Vps2 get cooperatively recruited by Snf7 (*Babst et al., 2002b*; *Mierzwa et al., 2017*; *Adell et al., 2017*), the same effect should be observed with overexpressed Vps2 as well, as the bound fraction of the complex of Vps24/Vps2 would increase upon overexpression of one of the components (*Figure 6C*).

We expressed Vps24 and Vps2 under two different expression systems (*Figure 6—figure supplement 1A*). Over-expressing Vps24 or Vps2 by ~2 fold (using a CEN plasmid) did not rescue the defect of the Snf7 helix-4 mutant. However, overexpressing Vps24 or Vps24 by ~16 fold (with a CMV promoter under a tet-off operator) completely rescued the *snf7*$^{D131K}$ defect (*Figure 6A–B*).

These analyses suggest that the overexpression of Vps24 or Vps2 increases the bound fraction of the Vps24/Vps2 complex in the Snf7$^{D131K}$ polymer, rescuing the defect of the mutant Snf7. Our in-vitro EM data suggest that the complex of Vps24/Vps2 interacts laterally on the Snf7 polymer. The polymer surface of Snf7 would provide multiple binding sites to the complex of Vps24/Vps2. Under overexpression conditions of Vps24/Vps2, there would be an increase in the bound fraction through avidity effects of Vps24/Vps2 to the Snf7$^{D131K}$ polymer.

The rescue by overexpression is less likely if Vps24/Vps2 were to bind at the end of the polymer, with a lower number of binding sites available to Vps24/Vps2. Consistent with this idea, overexpression of Vps24 or Vps2 did not rescue several longitudinal polymerization defective mutants of Snf7 (*Figure 6—figure supplement 1B–D*).

Altogether, our data provide strong evidence that Vps24 recognizes helix-4 in Snf7, which, along with Vps2, lies at the periphery of the core polymer, allowing for lateral association of these ESCRT-III proteins. This lateral interaction is mediated by promiscuous electrostatics, rather than specific residue-to-residue association.

## Discussion

The core components of the ESCRT-III complex are essential for numerous biological reactions. Deletions or mutations in Vps24 and or Vps2 are defective for cargo sorting through the MVB pathway. Despite the importance of these ESCRT-III proteins, the mechanism behind how they associate with other ESCRTs, in particular Snf7, remains undefined. How heteropolymerization of these proteins creates polymers of different curvatures and architectures remains uncharacterized. The molecular details that allow co-assembly of ESCRT-III proteins in general remains an important mechanistic question in cell biology.

Recent structural studies have provided important clues on the mechanism of activation and assembly of the mammalian ESCRT-III subunits IST1 and CHMP1B, the yeast ESCRT-III subunit Snf7, and the fly Snf7 ortholog Shrub. These structures defined the active, open conformation of the ESCRT-III proteins, and in the case of CHMP1B/IST1 copolymer, an open and a closed conformation. Previous genetic and biochemical experiments had suggested that different stimuli – membrane binding and ESCRT-II interaction – trigger 'opening' of the closed ESCRT-III conformation. This

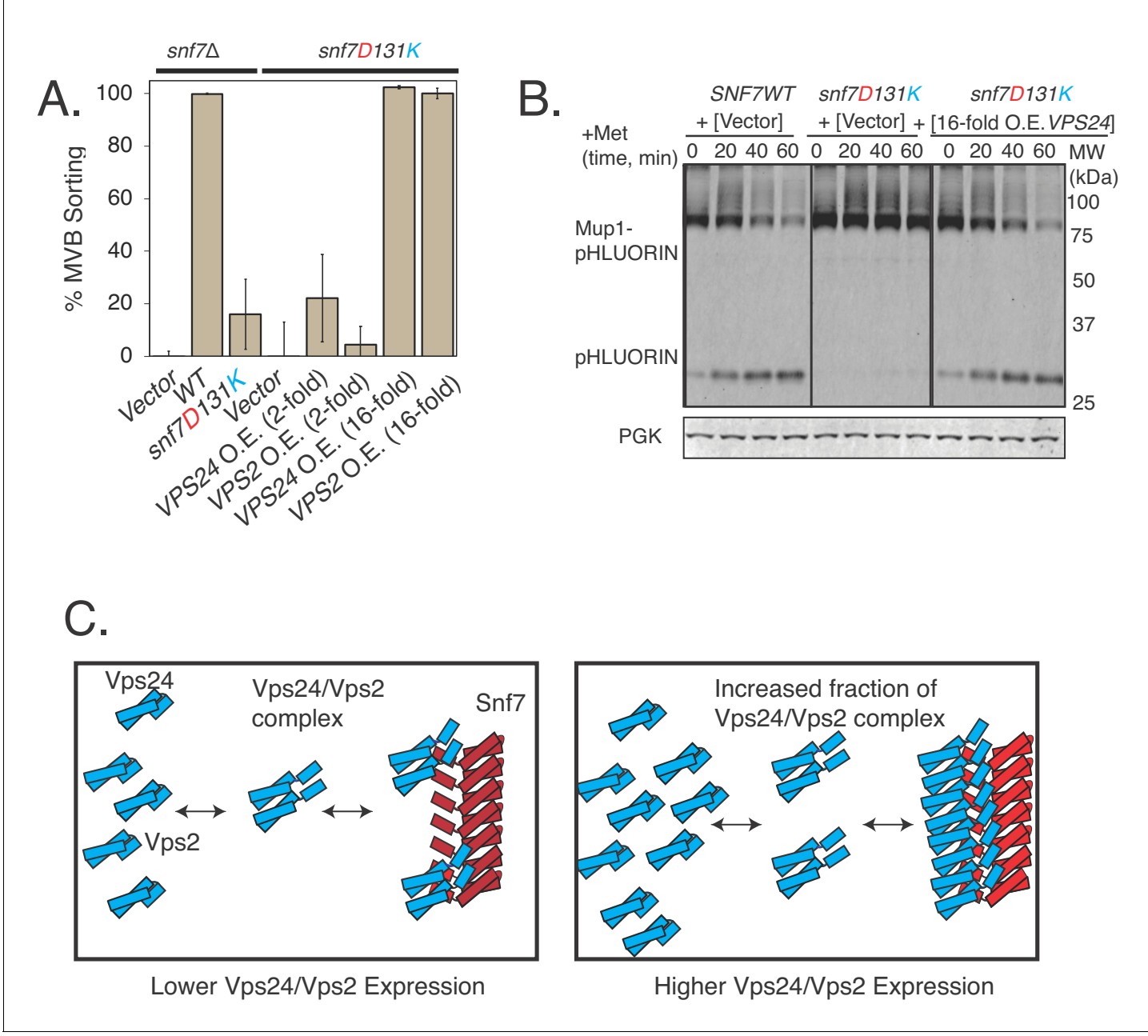

**Figure 6.** Vps24 and Vps2 cooperatively bind Snf7, as over-expressing them suppresses the defects of Snf7 helix-4 mutants. (A) Mup1-pHluorin endocytosis data after 90 min of methionine addition, showing overexpression of Vps24 or Vps2 rescuing the defect of the helix-4 mutant D131K. 2-fold overexpression (O.E.) corresponds to expressing either Vps24 or Vps2 with a CEN plasmid, while ~16 fold represents experiments performed with these genes on a CMV promoter with a tet-off operator. Error bars represent standard deviation from three to five independent experiments. (B). Overexpression of Vps24 (on a plasmid with a tet-off operator), rescues Mup1-pHluorin degradation defect of the D131K mutant, as shown immunoblotting of pHluorin over 60 min of methionine addition. (C).Model showing the hypothesized recruitment of Vps24/Vps2 in helix-4 mutant under normal (left) or under overexpressed (right) Vps24/Vps2 conditions.
DOI: https://doi.org/10.7554/eLife.46207.027

The following source data and figure supplement are available for figure 6:

**Source data 1.** Individual data points for the % MVB sorting of Snf7 mutant and suppression through overexpression of Vps24 and Vps2.
DOI: https://doi.org/10.7554/eLife.46207.029

**Figure supplement 1.** Overexpression of Vps24 or Vps2 do not suppress the defects of longitudinal polymerization mutants of Snf7.
DOI: https://doi.org/10.7554/eLife.46207.028

conformational change presents a hydrophobic surface and an electrostatic surface (*Tang et al., 2015*; *McMillan et al., 2016*) for the core Snf7 subunits to assemble into polymeric structures.

Here we find that the overall electrostatics of one of the helices of Snf7 (helix-4) is critical for Snf7 to recruit its partner Vps24. Mutations in this region of Snf7 inhibits degradation of model cargos because of the inability of these mutants to recruit Vps24.

Helix-4 of Snf7 is involved in lateral interaction of ESCRT-III subunits. In vitro, in the absence of another ESCRT-III partner, Snf7 uses this interface to assemble laterally, creating a ~ 9 nm protofilament, but it also can make additional contacts to produce filament bundles (*Chiaruttini et al., 2015*). We observe strong dependence of kinetics and thermodynamics in the formation of laterally interacting polymers, suggesting that over time and with an increase in concentration of the polymerizing species, Snf7 can progressively associate into lateral bundles. In the presence of another ESCRT-III subunit such as Vps24, this interface recruits the partner Vps24 to create a co-polymer, which then remodels the Snf7 spirals to create 3D helices through sliding of laterally interacting polymers (*Figure 7A*).

Recent evidence suggests that in vivo, the recruitment kinetics of ESCRT-III proteins Snf7, Vps24 and Vps2 is indistinguishable (*Mierzwa et al., 2017*; *Adell et al., 2017*). Therefore, pure polymers of Snf7, Vps24 or other ESCRT-III proteins, while providing important physical insights into polymerization dynamics, do not function individually in vivo. The function of the co-assembled heteropolymer is more relevant for membrane deformation.

Charge-inversion mutations in helix-1 of Vps24 rescue the defects exhibited by different Snf7 helix-4 mutants (*Figure 7A*). It is possible that these mutations in Vps24 allosterically enhance the affinity of Vps24/Vps2 for Snf7 rather than providing a direct interaction surface for Snf7. However, the following lines of evidence suggest that these mutants probably lie at the interface of Snf7/ Vps24. First, we find strong evidence that Snf7 helix-4 uses its acidic nature to recruit Vps24. Interestingly, helix-1 of Vps24 is predominantly basic (*Figure 3A*), and helix-1 of all ESCRT-III proteins is the most basic surface of these complexes. Second, the D131 (in helix-4) residue in the Snf7 crystal lattice contacts helix-1 in trans of a laterally coexisting polymer. Third, in thiol-based ex vivo crosslinking experiments, placing cysteines at Snf7 helix-4 and Vps24 helix-1 induces crosslinks. Fourth, in the cryo-EM structure of CHMP1B-IST1, CHMP1B helix-4 contacts the helix-1 in IST1 (*McCullough et al., 2015*). These data are consistent with the idea that Snf7 helix-4 contacts helix-1 of Vps24.

Interestingly, we observe that the overall electrostatics on one surface of helix-4 in Snf7 is important for this recognition, rather than a consensus sequence on that surface. The suppressor mutations in Vps24 lie in the basic helix-1 region. We note that our data point to the R19/K26 region of Vps24 as the binding surface for Snf7, and do not directly show that the binding surface may be spread out throughout the basic helix-1 region. However, it is possible that on a polymeric surface of Vps24 (and Vps2) where Snf7 is bound, each Snf7 monomer may engage with the same location of Vps24 (**R19**) through different residues of helix-4 (D127, D131, E142) (*Figure 7A*). This continuous charge-charge interaction among the polymers of ESCRT-III may be an important aspect of ESCRT-III co-assembly (*McCullough et al., 2015*; *McCullough et al., 2018*). On the one hand, multiplication of electrostatic interactions among each protomer would enhance the avidity of Snf7 to Vps24/Vps2 in the context of the polymer (*Figure 7B*). On the other hand, the uninterrupted presentation of charges as a binding surface would allow Vps24/Vps2 to adopt different positions along the filament (*Figure 7A–B*, *Figure 8A*). As the polymer constricts, the lack of a requirement for residue-to-residue specificity would enable Vps24/Vps2 to bind at different locations, allowing the polymers to adapt, by sliding side-by-side, to different curvatures in the polymer (*Figure 7A–C*, *Figure 8A*).

In the possible scenario that the dynamics of Snf7-Snf7 assembly and Vps24/Vps2 assembly are different (i.e. the on and off rates of Snf7 to the polymer is different from that of Vps24/Vps2) (*Chiaruttini and Roux, 2017*), the heteropolymer would further be able to embrace variable architectures capable of encircling cargo. Additionally, our crystal structure suggests that helix-4, and most likely the C-terminal region, which lie at the periphery of the polymer and away from it, can also exist in different conformations (*Tang et al., 2015*). An ability to easily change conformations at the periphery of the polymer could additionally enable the polymer to change its architecture.

Our data and the reported structure of CHMP1B/IST1 suggest that, similar to the core longitudinal assembly mechanism that are similar between these ESCRT-III proteins, the recognition of partners through helix-4 or the C-terminal peripheral region could be a general feature in ESCRT-III

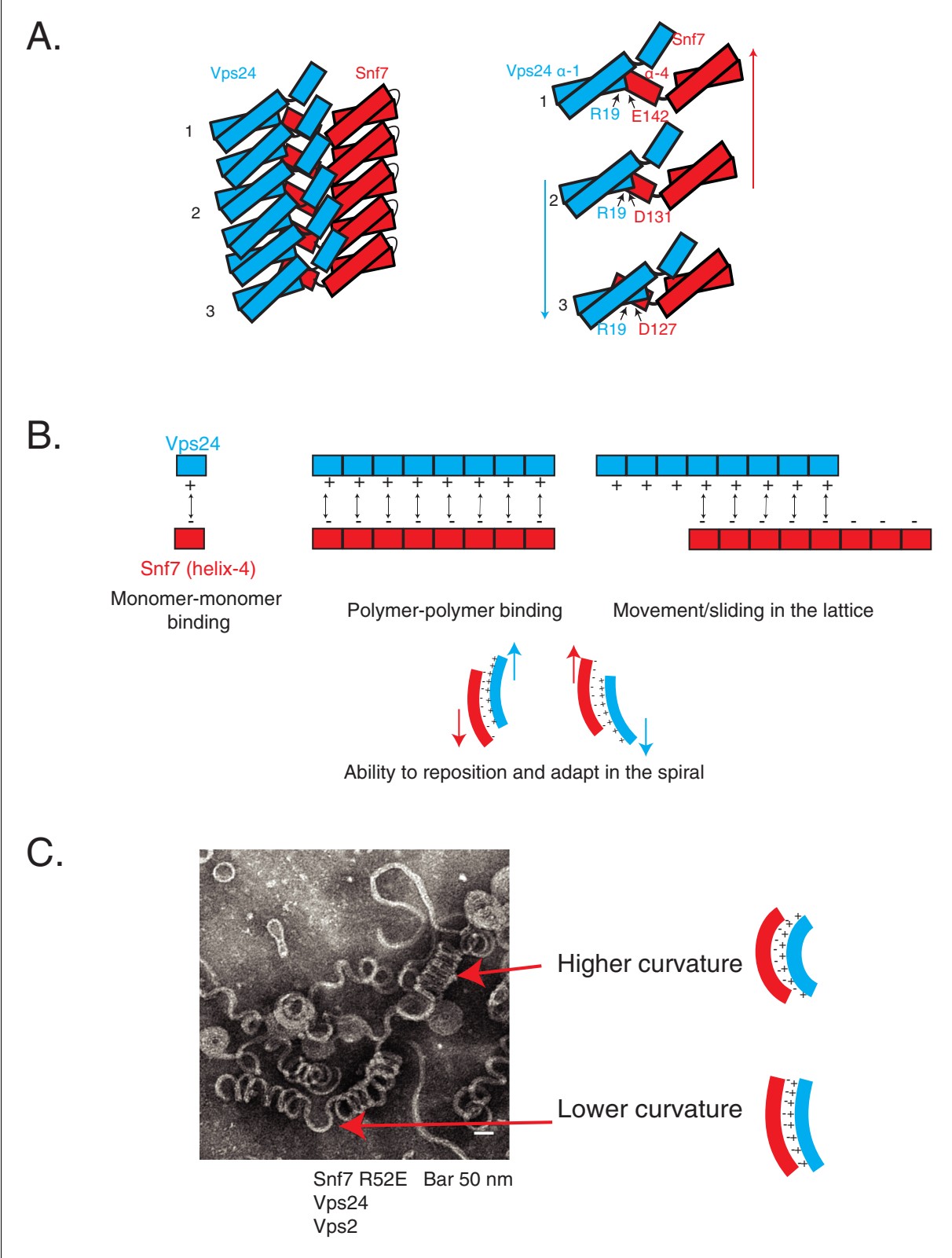

**Figure 7.** Cooperative lateral associations define ESCRT-III polymers. (**A**) Cartoon model of Snf7 filament interacting with Vps24 filament in trans in lateral fashion. On right, as an example, residue R19 of Vps24 is shown to interact with three different acidic residues of Snf7's helix-4, indicating its different positions along the polymer. Arrows represent possible sliding of the side-by-side polymers. (**B**) Model depicting laterally interacting polymers, in which multiple electrostatic interaction sites are present between the polymers, with the binding sites devoid of a specific registry between

*Figure 7 continued on next page*

*Figure 7 continued*

the residues involved in the interaction. The non-specific nature of the interaction could enable the polymers to slide along one another (top-right) and reposition and adapt (bottom) in the copolymer. (C) Electron microscopy image of 3D helices of Snf7 R52E, Vps24 and Vps2. Two positions are highlighted with arrows and corresponding models of the laterally interacting/sliding polymers are depicted on the right.

DOI: https://doi.org/10.7554/eLife.46207.030

assembly. It is important to note that CHMP1B also contacts IST1 through the C-terminal regions of CHMP1B (*Talledge et al., 2019*). In the case of the Snf7-Vps24 interaction, we find that Vps24-GFP is recruited to endosomes even with a Snf7 construct in which the C-terminus (helices-5 and beyond) is deleted (*Figure 2—figure supplement 2A*; *Henne et al., 2012*). This could mean that the C-terminus of Snf7 contacts Vps24 only with a weak affinity and therefore we are unable to see an obvious effect in our cellular assays. It is also possible that the C-terminus of Snf7 does not contact Vps24. Since the topology of membrane vesicles created by the reported CHMP1B-IST1 copolymer and those created by Snf7-Vps24/Vps2 at endosomes are opposite to one another, further analyses of these two systems is necessary for us to fully understand the similarities and differences between them. One possibility on how these two systems could create opposite topologies while possessing similar heteropolymer contacts is provided in *Figure 8—figure supplement 1*. In this model, while CHMP1B is bound to the membrane and resides inside of the polymer helix, Snf7 could reside on the outside, with the membrane bound to Snf7.

The combination of this plethora of different data is consistent with the following model: upon assembly of early ESCRTs (0, I and II) at the endosomes with cargo, Vps20 is recruited, which nucleates Snf7 polymerization. Snf7 forms a scaffold at the membrane which then is required to recruit Vps24/Vps2, although kinetically this recruitment likely happens simultaneously with Snf7 nucleation (*Figure 8B*).

Interestingly, a recently reported study of the bacterial tubulin homolog FtsZ suggests that weak van der Waals interactions between laterally interacting filaments may be an important property of that polymeric system (*Guan et al., 2018*). Furthermore, previous analyses suggested that partners of FtsZ can bundle FtsZ through lateral associations, giving rise to helical structures (*Goley et al., 2010*). Filament sliding models for FtsZ and its partners have been proposed before (*Szwedziak et al., 2014*). However, as far as we are aware, the molecular mechanisms of how such sliding may be accomplished by the filaments have not been defined. Given the corollary between ESCRT-III and FtsZ proteins (membrane associated spiraling polymers) such lateral associations created by non-specific interactions maybe a general way for such polymeric proteins to adopt different curvatures during assembly on malleable membrane surfaces (*Figure 8A*).

Although we observe that non-specific charge-charge interactions is a property of helix-4 in Snf7, we suspect such promiscuous interactions to be present within other interaction surfaces (in the longitudinal interface) as well. We and others have observed heterogeneity in the diameter of Snf7 spirals, suggesting that ESCRT-III polymers can adopt different diameters to be able to organize cargo inside the spirals. A rigid polymer with the same interaction surface will be unable to perform this task. As the inter-subunit angle changes and the polymers constrict, the contacts among the surface residues should also change to allow for generation of curvature.

Currently, structural analyses only depict an energetically stable complex with the same interaction surfaces in different lattices and are unable to clarify the dynamic ensemble of different interfaces. More structures of individual ESCRT-III proteins and in protein complexes will help clarify the interfaces that are important for homo and hetero polymerization of these fascinating self-assembling proteins and will allow us to understand how nature controls these ensembles.

## Materials and methods

### Yeast strains, plasmids and reagents

Strains, plasmids and reagents used in this study are listed in the Key Resources table. Previously used strains, plasmids, reagents were from the following references (*Robinson et al., 1988*; *Adell et al., 2017*; *Henne et al., 2012*; *Ghazi-Tabatabai et al., 2008*; *Tang et al., 2015*;

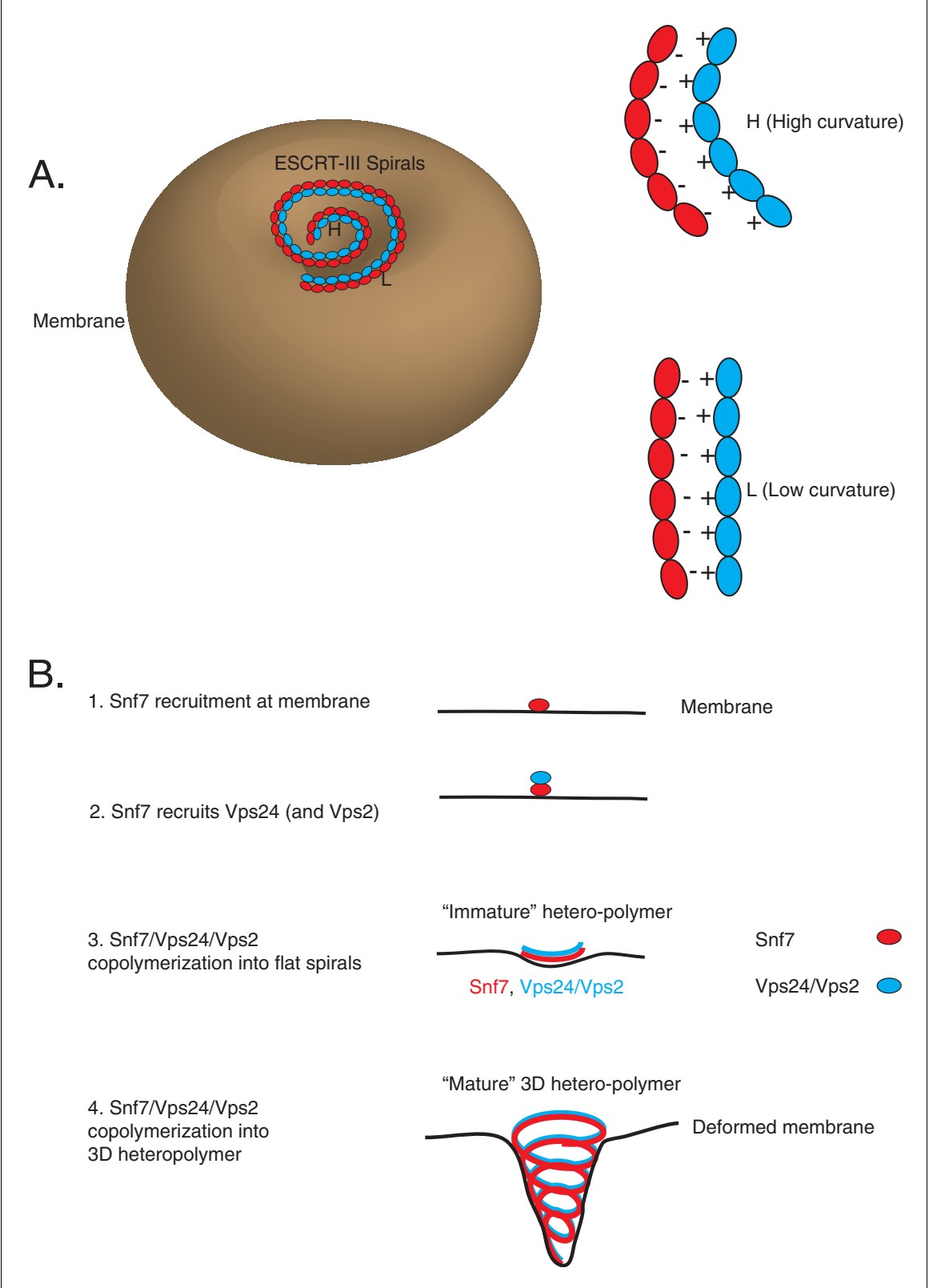

**Figure 8.** Co-assembly of ESCRT-III proteins into membrane-deforming polymers drive budding reactions. (A) Figure on the left depicts a model of ESCRT-III spirals budding membranes by utilizing different curvatures. Figure on the right depicts how laterally associating polymers can reposition themselves utilizing electrostatics at different curvatures. In the figures, H represent a position of high curvature and L represents a position of low

*Figure 8 continued on next page*

*Figure 8 continued*
curvature. (B) Model depicting different steps of the simultaneous recruitment of Snf7/Vps24/Vps2, which form spirals that drive membrane budding reactions.
DOI: https://doi.org/10.7554/eLife.46207.031
The following figure supplement is available for figure 8:

**Figure supplement 1.** Comparison of inside-out and outside-in tubulation properties of ESCRT-III proteins.
DOI: https://doi.org/10.7554/eLife.46207.032

*Tang et al., 2016*; *Buchkovich et al., 2013*; *Sikorski and Hieter, 1989*; *Garí et al., 1997*; *Babst et al., 1998*) and are also listed and referenced to in the table.

### Canavanine-sensitivity assays

Canavanine-sensitivity spot plating assays were performed as described before (*Lin et al., 2008*). Mid-log cells were diluted back to an optical density (at 600 nm) of 0.1. 10-fold serial dilutions were made and the dilutions applied to plates with selective drop-out media with different concentrations of canavanine. Images of the plates were taken after three or more days. Experiments were performed at least twice in all cases.

### Random mutagenesis

Canavanine resistance was used to select for mutations in Snf7 or Vps24 that function as suppressors of helix-4 mutant. To select for Vps24 mutants, a strain harboring a chromosomally integrated *snf7D131K* and *vps24Δ* was transformed with a plasmid library of randomly mutagenized *VPS24*. Random mutation was performed by error-prone PCR, as previously described (*Tang et al., 2016*), using primers annealing to the 5' and 3' ends of the *VPS24* ORF. Suppressing mutations on Snf7 were similarly obtained, mutagenizing the whole ORF of *snf7D131K* on a plasmid and then transforming to a *snf7Δ* strain.

### Mup1-pHluorin flow cytometry and immunoblotting

Flow cytometry analysis of Mup1-pHluorin endocytosis and trafficking was performed as described (*Henne et al., 2012*). Briefly, mid-log cells were treated with 20 μg/mL of L-methionine for 90 min. Cells were spun down and resuspended in synthetic dextrose complete minimal medium (SCD). Mean fluorescence of 100,000 cells were recorded using a BD Accuri C6 Flow Cytometer. % MVB sorting was calculated by normalizing the WT sorting to 100% and mutant (ESCRT deletion) to 0%. At least three independent experiments were performed to calculated standard deviation.

Western blots of Mup1-pHluorin was performed as follows. 5 OD equivalent of cells treated with 20 μg/mL methionine were collected by centrifugation at different time points at 4000 xg. Centrifuged cells were then washed with 1 mL of cold $H_2O$, and then centrifuged again at 4000 xg. Cells were then precipitated with 10% TCA for >1 hr on ice. Cells were washed twice with 1 mL of acetone, resuspending pellets between washes by bath sonication. Pelleted cells were then lysed in 100 μL lysis buffer (50 mM Tris-HCl, pH 7.5, 8 M urea, 2% SDS, and 1 mM EDTA) by bead beating for 10 min. 100 μL of sample buffer (150 mM Tris-Cl, pH 6.8, 8 M urea, 10% SDS, 24% glycerol, 10% v/v βME, and bromophenol blue) was then added to the sample and vortexed for 10 min. After centrifugation for 6 min at 21,000 xg, supernatant was loaded on an SDS-PAGE gel and transferred onto a nitrocellulose membrane. Rabbit polyclonal GFP antibody (Torrey Pines) was used to detect pHluorin. Imaging of the western blots was performed using an Odyssey CLx imaging system and analyzed using the Image Studio Lite 4.0.21 software (LI-COR Biosciences).

### Sequence alignment and structural analyses

Snf7 sequences were aligned using Mafft (*Katoh et al., 2002*). Jalview (*Clamp et al., 2004*) was used to visualize the sequences. Homology modeling of the Vps24 structure was performed using Modeller (*Fiser and Sali, 2003*), using the CHMP3 (PDB 3FRT) structure as the template. Helical wheel analysis was performed using Heliquest (*Gautier et al., 2008*). Structures were viewed and analyzed using UCSF Chimera (*Pettersen et al., 2004*).

## Fluorescence microscopy

1 mL of mid-log cells expressing *VPS24-GFP* were centrifuged for 2 min at 10,000 xg, and resuspended in 25 mL of synthetic media. Microscopy was performed on a Deltavision Elite system with an Olympus IX-71 inverted microscope, using a 100X/1.4 NA oil objective. Image extraction and analysis were performed using the FiJi software (*Schindelin et al., 2012*).

## Subcellular fractionation

15 ODs of mid-log cells were harvested and spheroplasted using zymolyase treatment as previously described (*Buchkovich et al., 2013*). Spheroplasts were lysed by douncing on ice in 50 mM Tris pH 7.5, 200 mM sorbitol with protease inhibitors. Lysates were centrifuged at 500 xg at 4°C. This supernatant (S5) was then centrifuged at 13,000 xg for 10 min at 4°C, which provided us P13 (endosome enriched pellet fraction at 13,000 xg) and S13 (supernatant fraction). The P13 and S13 fractions were then precipitated in TCA, and immunoblotted as described above for Mup1-pHluorin.

## Co-immuoprecipitation

30 ODs of mid-log cells were harvested and spheroplasted as done for subcellular fractionation experiments. Lysis was performed by douncing in 50 mM Hepes pH 7.5, 200 mM Sorbitol, 150 mM NaCl, 1 mM EDTA, 1 mM DTT and 1%-TritonX-100. Lysate was then centrifuged at 13,000 xg. Supernatant was then treated with protein G beads (Dynabeads) for 30 min at 4°C to clear background binding to beads. After centrifugation at 500 xg for 10 min, supernatant was then incubated with anti-Vps24 antibody (at 1/250 dilution) for 2 hr. Protein G beads were then used to pull-down Vps24 bound complexes. After washing three times with PBS buffer at 20 fold excess volume of the beads, the beads were treated with sample buffer (150 mM Tris-Cl, pH 6.8, 8 M urea, 10% SDS, 24% glycerol, 10% v/v βME, and bromophenol blue). After SDS-PAGE, western blots were performed, and anti-Snf7 and anti-Vps24 antibodies were used to probe bound complexes.

## Crosslinking

30 ODs of cells were harvested in 50 mM Hepes pH 7.5, 200 mM Sorbitol, 150 mM NaCl, 1 mM EDTA, fresh 0.5 mM DTT and Roche's complete protease inhibitor. Lysis was performed by bead-beating (Zirconia-Silicon beads) twice for 30 s, with 30 s intervals on ice. After lysis, the lysate was supplemented with 1% of Triton-X 100 and incubated at 4°C for 20 min. Lysate was cleared by centrifugation at 500 xg for 5 min at 4°C. Supernatant was treated with 3.3 mM of BMOE (bismaleimido-ethane) and 10 mM EDTA and incubated at 4°C for 10 min. Reaction was stopped using 10 mM of DTT and the solution then treated with 10% of TCA. TCA precipitation was performed for >1 hr. Western blots were performed as described above.

## Glycerol gradient fractionation

30 ODs of cells were harvested in PBS buffer, fresh 0.5 mM DTT and Roche's complete protease inhibitor. Lysis was performed by bead-beating (Zirconia-Silicon beads). Glycerol gradients were made using Gradient Master 108 from Biocomp. Centrifugation was performed at 100,000 xg for 4 hr at 4°C. 1 mL fractions were collected from the solutions, TCA precipitated and immunoblotted as described above.

## Protein expression and purification

Snf7, Vps24 and Vps2 and Vps20 proteins were expressed from a modified pET28a(+) vector expressing His6-SUMO protein at the N-terminus. GST-Vps25 was expressed using the pGEX6p1 vector. Expression of ESCRT-III was performed using the Rosetta *E. coli* strain. Snf7, Vps20, Vps24 and GST-Vps25 were constructs were expressed at 37°C for 4 hr by inducing with 0.5 mM IPTG. Snf7-D131K and Vps2 were expressed at 26°C overnight, inducing with 0.5 mM IPTG.

Harvested cells were lysed by sonication. Affinity purification of the proteins through the His6 tag was performed using $Co^{2+}$ talon resin. The SUMO tag was cleaved overnight at 4°C on the beads using ULP1 protease. Eluate was subjected to a Hi-trap Q Seph FF column. The anion exchange eluate was concentrated and ran through a Superdex 200increase column. GST-Vps25 was purified using GSH-sepharose beads, eluted using glutathione, concentrated and ran through a Superdex 200 column. Eluted proteins were concentrated, flash-frozen in liquid nitrogen and stored at −80°C.

## Liposome sedimentation

Liposomes were made using a mixture of 60% POPC and 40% POPS. Lipids in chloroform were mixed at the appropriate molar ratios and dried overnight under vacuum. Lipids were hydrated for 3 hr and resuspended in 25 mM Hepes 7.5, 150 mM NaCl, to make a lipid concentration of 1 mg/mL. Large unilamellar vesicles (LUVs) were made using extrusion filters of 800 nm pores from Avanti Polar Lipids. Proteins were added to the liposomes at a final concentration of 200 nM and a final lipid concentration of 0.5 mg/mL. After incubation for 30 min at room temperature, centrifugation was performed in a TLA-100 (Beckman Coulter) at 70,000 rpm for 10 min at 4°C. SDS-PAGE was used to determine fraction of protein pelleted with the liposomes.

## Lipid monolayers formation and electron microscopy

Lipid monolayers were formed using a ratio of 60% POPC, 30% POPS and 10% PI3P in chloroform. Monolayers were formed above an aqueous buffer solution, and lipids were injected underneath the monolayer, using a home-made Teflon apparatus, as described before (*Henne et al., 2012*). Carbon-coated electron microscope grids were applied to the top of the aqueous solution simultaneously with the application of proteins. Incubation of proteins on the monolayers/grids were performed at various times as indicated in the text. Grids were then stained with 2% ammonium molybdate and imaged on FEI Morgagni 268 TEM.

## Acknowledgements

We thank David Teis for the kind gift of strain harboring *Vps24-LAP-eGFP*. We thank Mike Henne, Richa Sardana, Jeff Jorgensen and Sho Suzuki for critical reading of the manuscript, all members of the Emr lab, Chris Fromme, Yuxin Mao and Peter Borbat for discussions. Work in the Emr lab is supported by a Cornell University Research Grant CU3704. Sudeep Banjade is a HHMI fellow of the Damon Runyon Cancer Research Foundation (DRG-2273–16). Shaogeng Tang is a Merck fellow of the Damon Runyon Cancer Research Foundation (DRG-2301–17) on a separate project.

## Additional information

### Funding

| Funder | Grant reference number | Author |
|---|---|---|
| Cornell University | CU3704 | Scott D Emr |
| Damon Runyon Cancer Research Foundation | DRG-2273-16 | Sudeep Banjade |
| National Institute of General Medical Sciences | T32GM007273 | Shaogeng Tang |

The funders had no role in study design, data collection and interpretation, or the decision to submit the work for publication.

### Author contributions

Sudeep Banjade, Conceptualization, Data curation, Formal analysis, Funding acquisition, Validation, Investigation, Methodology, Writing—original draft, Writing—review and editing; Shaogeng Tang, Conceptualization, Formal analysis, Investigation, Methodology, Writing—review and editing; Yousuf H Shah, Investigation, Writing—review and editing; Scott D Emr, Conceptualization, Resources, Supervision, Funding acquisition, Writing—review and editing

### Author ORCIDs

Sudeep Banjade (ID) https://orcid.org/0000-0002-5920-891X
Shaogeng Tang (ID) https://orcid.org/0000-0002-3904-492X
Scott D Emr (ID) https://orcid.org/0000-0002-5408-6781

Decision letter and Author response
Decision letter https://doi.org/10.7554/eLife.46207.036
Author response https://doi.org/10.7554/eLife.46207.037

## Additional files

### Supplementary files

• Supplementary file 1. Key resources table.
DOI: https://doi.org/10.7554/eLife.46207.033

• Transparent reporting form
DOI: https://doi.org/10.7554/eLife.46207.034

### Data availability

All data generated are included in the manuscript and supporting files.

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
