## [Decision Letter]

Thank you for submitting your article "Electrostatic lateral interactions drive ESCRT-III heteropolymer assembly" for consideration by *eLife*. Your article has been reviewed by three peer reviewers, one of whom is a member of our Board of Reviewing Editors, and the evaluation has been overseen by John Kuriyan as the Senior Editor. The following individual involved in review of your submission has also agreed to reveal his identity: Christopher P Hill (Reviewer #3).

The reviewers have discussed the reviews with one another and the Reviewing Editor has drafted this decision to help you prepare a revised submission.

Summary:

The reviewers agree that this work significantly advances understanding of ESCRT-III filament assembly. Considerable evidence is presented in support of a model in which the conserved acidic helix 4 of Snf7 contacts a basic helix (Helix 1) of Vps24 in a relatively non-specific manner, allowing for plasticity in the interactions between these components that gives rise to different filament architectures. These data include genetic reporters, endosomal sorting, suppressor screening, crosslinking, co-immunoprecipitation assays, and negative stain EM. There are no technical concerns, and the most fundamental conclusion, that interactions between Snf7 Helix 4 and Vps24 Helix 1 are functionally important, is well-supported.

Essential revisions:

1) The model proposing an electrostatically driven flexible interface between Snf7 Helix4 and Helix 1 of Vps24 is consistent with the highly variable ESCRT-III curvature observed in prior studies. Nevertheless, the reviewers felt that evidence presented for the model is not decisive in the absence of a more detailed structural model. The evidence for promiscuous interactions comes principally from the suppressors shown in Figure 4; less so for the charge inversion mutants in helix 1, which are all close by, possibly within one turn. Likewise, the equivalent residues of R19 and K26 in the CHMP3 structure are very close, so an 8 Å spacer in the crosslinker might react with both in a single conformation. The authors need to more carefully discuss the extent to which these data establish the model.

2) In addition to showing the inferred interactions mapped onto their earlier Snf7 crystal structure, it would be helpful if the authors could include a more complete discussion and illustration of how the interacting surfaces might map into the cryo-EM structure of IST1-CHMP1 that was published by Frost and colleagues. Are the interacting surfaces inferred from Snf7 crystal contacts and the from the IST1-CHMP1B cryo-EM structure the same as each other, and to what extent is each of them consistent with the current data? More complete discussion of this point, with relevant figures, will be of interest because of the dramatic conformational changes inferred for Snf7 in earlier studies, and conformational differences seen between IST1 and CHMP1B, and because of the very different pathways and membrane topology changes associated with these different ESCRT-III isoforms.

3) Please clarify what is meant by "mature polymerization".

4) Figure 5: Why does the presence of Vps20 prevent Snf7-Vps24-Vps2 helix formation?

---

## [Author Response]

Essential revisions:1) The model proposing an electrostatically driven flexible interface between Snf7 Helix4 and Helix 1 of Vps24 is consistent with the highly variable ESCRT-III curvature observed in prior studies. Nevertheless, the reviewers felt that evidence presented for the model is not decisive in the absence of a more detailed structural model. The evidence for promiscuous interactions comes principally from the suppressors shown in Figure 4; less so for the charge inversion mutants in helix 1, which are all close by, possibly within one turn. Likewise, the equivalent residues of R19 and K26 in the CHMP3 structure are very close, so an 8 Å spacer in the crosslinker might react with both in a single conformation. The authors need to more carefully discuss the extent to which these data establish the model.

We agree with the reviewers that our model would be more decisive in the presence of a structural model. The fact that the interaction between Snf7 and Vps24 is promiscuous and the polymer created are of variable architectures is possibly the reason we have been unable to crystallize the complex of a Snf7 molecule with truncated or full-length versions of Vps24. However, we do believe, as suggested by the reviewers, that our data with suppressor mutations in Snf7 strongly argue for a promiscuous interaction between Snf7 and Vps24. This latter point would have not been obvious from one static structure, and this is why we believe that our genetic and biochemical work will be of much interest and use to the ESCRT community and beyond.

We agree that the interaction site on Vps24 may be close to the R19/K26 region, and not very spread out on the monomer, although we do see that mutating Q16 or K33, which are quite far apart (~ 28 Angstroms in the homology model) still induces rescue of binding. However, on laterally interacting polymers of Snf7 and Vps24, even the same surface of Vps24 (R19/K26) would be multiplied across the polymer lattice for the acidic surface of Snf7 to interact with. Therefore, it is possible that with the same interaction site (R19/K26 surface), different residues of Snf7 helix-4 may interact across the polymer.

We have been purposefully cautious in explaining the interaction surface of Vps24, and we have added text (Discussion, sixth and seventh paragraphs; subsection “Basic to acidic mutations in helix-1 of Vps24 rescue the defect of Snf7 helix-4 mutants”, last paragraph) to explain the model better.

2) In addition to showing the inferred interactions mapped onto their earlier Snf7 crystal structure, it would be helpful if the authors could include a more complete discussion and illustration of how the interacting surfaces might map into the cryo-EM structure of IST1-CHMP1 that was published by Frost and colleagues. Are the interacting surfaces inferred from Snf7 crystal contacts and the from the IST1-CHMP1B cryo-EM structure the same as each other, and to what extent is each of them consistent with the current data? More complete discussion of this point, with relevant figures, will be of interest because of the dramatic conformational changes inferred for Snf7 in earlier studies, and conformational differences seen between IST1 and CHMP1B, and because of the very different pathways and membrane topology changes associated with these different ESCRT-III isoforms.

We thank the reviewers for suggesting these direct comparisons, which have improved the manuscript further. We have added text in the Results (subsection “Basic to acidic mutations in helix-1 of Vps24 rescue the defect of Snf7 helix-4 mutants”, fourth paragraph), Discussion (ninth paragraph), and added two figures (Figure 2—figure supplement 3, and Figure 8—figure supplement 1) for these direct comparisons.

3) Please clarify what is meant by "mature polymerization".

We find strong dependence of both concentration and time in our in vitro polymerization assays for the formation of heteropolymers of ESCRT-III. The use of the term “mature polymerization” was to suggest the idea that the polymers have reached their membrane-budding abilities (probably helices of an unclear length) over time and in their activated stage. In the absence of a clear idea of what the final membrane budding polymer looks like, we use the term “mature.” We have changed the language in the fifth paragraph of the subsection “Lateral cooperative interactions mediated by electrostatics drive ESCRT-III co-assembly” and in the last paragraph of the subsection “Lateral cooperative interactions mediated by electrostatics drive ESCRT-III co-assembly”, to better explain these ideas.

4) Figure 5: Why does the presence of Vps20 prevent Snf7-Vps24-Vps2 helix formation?

The experiments with Vps20, Snf7, Vps24 and Vps2 were done in the presence of the wild-type Snf7, as opposed to the activating mutation R52E in Snf7. We find that the helix formation is more robust in the presence of the activating mutation, but absent in the presence of wild-type Snf7, even with Vps20 present, which is suggested to be the nucleator of Snf7 polymerization. With Vps20 and Snf7-R52E both present, we are still able to see bundled helices (new figure, Figure 5—figure supplement 4), suggesting that Vps20 in fact may not be inhibiting helix formation.

We believe that in our in-vitro assays Vps20 alone is not sufficient to fully activate Snf7. This is probably because the Vps20 we use in vitro is not myristoylated, as it is in vivo, causing it to be not fully recruited to the membrane. In the presence of a GST-tagged Vps25 (an ESCRT-II component), we start observing helices of Snf7-Vps24-Vps2 (new figure, Figure 5—figure supplement 4), suggesting that in our assays additional nucleators/activators are necessary to induce helix formation in the presence of the wild-type Snf7 protein. These explanations are also added to the text (subsection “Lateral cooperative interactions mediated by electrostatics drive ESCRT-III co-assembly”, sixth paragraph).